# Transcriptomic Harmonization as the Way for Suppressing Cross-Platform Bias and Batch Effect

**DOI:** 10.3390/biomedicines10092318

**Published:** 2022-09-18

**Authors:** Nicolas Borisov, Anton Buzdin

**Affiliations:** 1World-Class Research Center “Digital Biodesign and Personalized Healthcare”, Sechenov First Moscow State Medical University, 119435 Moscow, Russia; 2Moscow Institute of Physics and Technology, 141701 Dolgoprudny, Russia; 3Shemyakin-Ovchinnikov Institute of Bioorganic Chemistry, 117997 Moscow, Russia; 4PathoBiology Group, European Organization for Research and Treatment of Cancer (EORTC), 1200 Brussels, Belgium

**Keywords:** gene expression, transcriptional profiles, RNA sequencing, microarray hybridization, data normalization and harmonization, batch effect, machine learning, Big Data, universal data indexing

## Abstract

(1) Background: Emergence of methods interrogating gene expression at high throughput gave birth to quantitative transcriptomics, but also posed a question of inter-comparison of expression profiles obtained using different equipment and protocols and/or in different series of experiments. Addressing this issue is challenging, because all of the above variables can dramatically influence gene expression signals and, therefore, cause a plethora of peculiar features in the transcriptomic profiles. Millions of transcriptomic profiles were obtained and deposited in public databases of which the usefulness is however strongly limited due to the inter-comparison issues; (2) Methods: Dozens of methods and software packages that can be generally classified as either flexible or predefined format harmonizers have been proposed, but none has become to the date the gold standard for unification of this type of Big Data; (3) Results: However, recent developments evidence that platform/protocol/batch bias can be efficiently reduced not only for the comparisons of limited transcriptomic datasets. Instead, instruments were proposed for transforming gene expression profiles into the universal, uniformly shaped format that can support multiple inter-comparisons for reasonable calculation costs. This forms a basement for universal indexing of all or most of all types of RNA sequencing and microarray hybridization profiles; (4) Conclusions: In this paper, we attempted to overview the landscape of modern approaches and methods in transcriptomic harmonization and focused on the practical aspects of their application.

## 1. The Problem of Transcriptomic Data Harmonization 

The digital ocean of whole-transcriptome gene expression profiles has flooded since the early 2000s when the first generation of robust and reproducible mRNA microarray hybridization (MH) techniques was introduced into the routine laboratory practice [1,2,3,4]. The outstandingly high importance of the open-access gene expression data that could be accumulated and extracted from public databases was recognized immediately, thus leading to emergence of popular online repositories such as Gene Expression Omnibus (GEO) [5,6] or ArrayExpress [7,8]. Later on, this has also inspired many impactful large-scale integrative biomedical cooperative projects such as The Cancer Genome Atlas (TCGA) [9,10] for cancer genomics and transcriptomics, Gene-Tissue Expression (GTEx) [11,12], and Atlas of Normal Tissue Expression (ANTE) [13] for normal human tissue expression profiles, the CancerRxGene database [14] for genomes and transcriptomes of cell lines connected with their response to hundreds of drugs, and the Broad Institute deconvoluted profiles for gene expression changes in cells under the influence of gene constructs, drugs, and other chemicals [15,16].

Shortly after the critical mass of gene expression profiles has accumulated, the following two conceptual problems with the data analysis were recognized. First, poor technical compatibility of the expression profiles is obtained using different experimental platforms/equipment, protocols, and reagents [17,18,19,20,21]. Indeed, this can be readily explained by the different physico-chemical principles of gene detection and interrogation [22,23] and by specific library preparation enzymatic bias [24]. The second problem (so-called batch effect) dealt and still deals with the unclear compatibility of gene expression profiles obtained with the same equipment and reagents, but in different series of experiments, e.g., they are performed in different times or in different labs [25,26]. There is no clear explanation of the nature of the batch effect (e.g., it may be due to relatively different activities of enzymes and chemicals for library preparation and MH or RNA sequencing from batch to batch), but the effect itself is sound and frequently inevitable [25]. 

The compromised compatibility of gene expression profiles obtained using different platforms and protocols was experimentally explored in the international projects MAQC (for MH) and SEQC (for RNA sequencing). Both MAQC [17,18,19] and SEQC [27] projects investigated compatibilities of gene expression profiles obtained using various microarray and sequencing platforms for the same set of four sample types (named A, B, C, and D), each performed in multiple replicates. Type A samples were the commercially available Stratagene Universal Human Reference RNA specimens for all but brain human tissues; type B samples were also commercially available Ambion Human Brain Reference RNA. Types C and D samples were the mixtures of A and B with the A:B ratios of 3:1 and 1:3, respectively. In the MAQC project [17,18,19], the samples of types A, B, C, and D were profiled using the MH platforms Agilent-012391 Whole Human Genome Oligo Microarray G4112A (GPL1708), Affymetrix Human Genome U133 Plus 2.0 Array (GPL570) and Illumina Sentrix Human-6 Expression Beadchip (GPL2507). In the SEQC project [27], the same samples were profiled using the NGS platform Illumina HiSeq 2000 (GPL11154), as well as three MH platforms: Illumina HumanHT-12 V4.0 expression beadchip (GPL10558), Affymetrix Human Gene 2.0 ST Array (GPL17930), and Affymetrix GeneChip^®^ PrimeView™ Human Gene Expression Array (GPL16043). 

The MAQC and SEQC projects investigated the correlations between the transcriptome profiles of the same biological type, yet obtained using the different experimental platforms. Although these correlations were high [17,18,19,27], without the some special cross-platform normalization methods (quantile normalization [28] was not enough), the overall collections of profiles were grouped according to the experimental platforms, rather than to the biological type of samples, in terms of both clustering dendrograms and of principal component analysis (PCA) [29,30,31,32,33,34]. 

As the reaction of the scientific community, a bunch of first-generation harmonization/normalization methods was generated in the first decade of the 21st century, aimed at the standardization of multi-platform expression profiles using specific algorithms. These methods were mostly trained on the different types of MH gene expression data and could dynamically transform gene profiles into a flexible yet inter-comparable form [35]. The following alternative approaches that have different principles and different destinies could be mentioned in this review: Quantile Normalization (QN) [28], Quantile Discretization (QD) [36], Normalized Discretization (NorDi) [37], Distribution Transformation (DisTran) [38], Empirical Bayes (EB)/ComBat [39], Distance-Weighted Discrimination (DWD) [40,41,42], Cross-Platform Normalization (XPN) [29,31], Gene Quantiles (GQ) [43], and PLatform-Independent Latent Dirichlet Allocation (PLIDA) [30]. 

Further approaches were largely influenced by the coming era of routine next-generation sequencing (NGS) of mRNA (RNA sequencing or RNAseq) that has started roughly in the second decade of this century. Nowadays, RNAseq has become the gold standard and the basic tool for transcriptomic profiling [44,45,46,47,48,49,50]. In addition to measuring gene activities, RNAseq has also the potential of detecting mutations and overall tumor mutational burden [51], gene splice isoforms [52], and oncogenic fusion transcripts [53,54,55,56]. During the RNAseq era, a new group of cross-platform data comparison methods was developed [27]. However, the RNAseq gene expression profiles have outnumbered the MH counterparts relatively recently. It was only in 2019 [32] when the number of datasets for the most popular RNAseq platform (Illumina HiSeq 2000, GPL11154) exceeded the number of datasets for the most popular MH platform (Affymetrix U133, GPL570). Moreover, the total number of individual profiles for those two platforms is still comparable in 2022 as well. Many unique transcriptomic profiles exist only as the MH data, e.g., clinically annotated expression profiles for some pathological conditions including cancers [57].

For the RNAseq data, a method called The Differential gene Expression in Sequencing, DESeq [58]/DESeq2 [59,60,61] based on the negative binomial distribution law, has rapidly become the standard in the field for the intra-platform normalization. However, effective method for the cross-platform RNAseq, or for RNAseq vs. MH harmonization was missing until recently, although several attempts for simultaneous normalization of MH and RNAseq data must be mentioned, such as Training Distribution Machine (TDM) [62], Universal exPression Code (UPC) [63], Feature-specific QN (FCQN) [64,65], MatchMixeR (MM) [66], Integrative Bayesian Network (IBN) [67], Rank-in [68], and Elastic Shared LASSO Regularization (ESLR) [69] methods. The divergence analysis method is another interesting attempt to compare the MH and NGS mRNA expression profiles, as well as microRNA and DNA methylation data [70]. The authors of the divergence analysis first applied conditional probability (Bayesian) models to mimic the unspecified (generalized-type) distribution that describes the gene expression/methylation data. This reconstruction was followed by the divergence analysis of one biological sample type from another. Although Dinalankara at al. [70] have managed to distinguish different samples after their divergence analysis, the applicability of their approach to a wide range of popular MH and NGS platforms remains unexplored [70]. 

In this review, we classified available intra- and cross-platform harmonization methods of transcriptomic profiles and compared their performance characteristics. Finally, we also included practical recommendations that may guide the reader to select optimal method depending on a specific task. 

## 2. Principles of Harmonization Algorithms 

Different harmonization methods are based on different algorithms aimed to suppress the platform bias and the batch effect. These algorithms may utilize different approaches to gene expression data processing and produce output data in different formats. Considering the mathematical apparatus, we proposed the following classification:
(1)Methods based on statistical transformations (considering quantiles, ranks, means, medians of gene expression levels, etc.):
(a)Those using ranking of expression levels and setting the output levels according to the averaged values, such as QN [28], Feature-Specific QN (FCQN) [64], Quantile Discretization (QD) [36], Gene Quantiles (GQ) [43], Normalized Discretization (NorDi) [37], Distribution Transformation (DisTran) [36,38], Median Rank Scores (MRS) [36], YuGene [71], and Rank-in [68];(b)Those using piecewise rescaling of log-expression levels according to the mean/median values over distinct genes and samples, such as Column Sample (CS), Median-Centered (MC) [29], and Analysis of Variance (ANNOVA) [72] method;
(2)Methods using regression and/or maximum likelihood models for validation of predefined statistical hypotheses:
(a)Those using negative binomial distribution, such as the DESeq [58]/DESeq2 [59,60,61];(b)Those using log-normal distribution with either covariance analysis [73], or with conditional/Bayesian models, as for the methods Universal exPression Code (UPC) [63,74], Empirical Bayes (ComBat) [39], Robust Microarray Analysis (RMA) [75], GeneChip Robust Multiarray Analysis (gcRMA) [76], Model-Based Expression Indices (MBEI) [77], Probe Logarithmic Intensity ERror (PLIER) estimation [78], frozen Robust Microarray Analysis (fRMA) [79,80,81,82], MatchMixeR (MM) [66], Cross-Platform Comparison (XPC) [83];(c)Those using Dirichlet and gamma distributions as for the method PLatform-Independent Latent Dirichlet Allocation (PLIDA) [30];(d)Those using the empirical superposition of conditional probabilistic (Bayesian) models that describe the generalized-type distribution as for the method applied for the comparison of the MH, NGS, microRNA, and DNA methylation data [67,70];(e)Those using the Least Absolute Shrinkage and Selection Operator (LASSO) regression models [69];
(3)Methods finding similar clusters in gene expression matrices of the datasets under normalization and then using iterative corrections to fit each cluster as close as possible to the target model:
(a)Those using piecewise linear interpolations in the log-expression space, such as Cross-Platform Normalization (XPN) [29];(b)Those using piecewise cubic interpolations in the log-expression space, such CuBlock [34].
(4)Methods utilizing machine learning (ML) to find and artificially remove dissimilarities between datasets to be normalized:
(a)Those using the linear support vector machine (SVM) ML method, such as Distance-Weighted Discrimination (DWD) [40,41,42];(b)Those using quantile-based regression models for data transfer from source to target datasets, such as Training Distribution Machine (TDM) [62].



Another important aspect that must be considered in this review is the format of output gene expression data generated by the harmonization techniques. Most of currently existing methods return the results in the flexible format. For the flexible normalization, the shape of the output transformed gene expression profiles is a variable that depends on all the profiles under harmonization. This has an important limitation that one cannot combine the output datasets generated after two or more acts of such harmonization. Even adding as few as just one transcriptional profile would require a new harmonization of the entire dataset. This clearly increases the calculation costs for large datasets that are being routinely updated.

Taken together, these factors complicate the analysis of not only single gene expression levels, but also of higher order gene-based biomarkers such as gene signatures [84], molecular pathway activation levels [85], algorithmically deduced cancer drug efficiency scores [86,87], and different ML models [88,89,90].

To overcome these limitations, an alternative concept was formulated comprising conversion of a whole set of profiles under harmonization into a pre-defined output shape, e.g., into a shape of a preferred gene interrogating experimental platform. In such a paradigm, the harmonized output should look as if it would be obtained using a predefined gene expression platform. The examples of predefined-shape harmonization methods include Frozen Robust Microarray Analysis (fRMA) [79,80,81,82], robust Quantile Normalization [91], Training Distribution Machine (TDM) [62], and Universal exPression Code (UPC) [63].

More recently, we proposed a new family of uniformly shaped cross-platform harmonizers termed Shambhala [32,33]. Harmonization here is performed not simultaneously for all the profiles under harmonization, but for the gene expression profiles taken one by one, when each individual profile is merged and quantile-normalized [28] with an auxiliary calibration dataset that is pre-defined by the method developers. Then, the resulting dataset is converted into the shape of the so-called reference definitive dataset. This creates an additional advantage of co-harmonizing datasets of different, even non-comparable, sizes.

Furthermore, such harmonization may use different mathematical transforms as the engine to reshape the transcriptional profiles. The first version of Shambhala used the piecewise linear method XPN [29,31] for profile reshaping [32], whereas the latest version [33] utilized the piecewise cubic transformation method CuBlock [34]. 

## 3. Evaluation of the Quality of Harmonization 

Harmonization of transcriptional profiles is a complex process that can distort functionally relevant features such as clustering and neighborhood on a dendrogram and fold-change of gene expression with relation to control samples. We listed in Table 1 some of the quality assessment metrics and the abilities of different methods to retain the initial functional characteristics in the output profiles after harmonization.

The following quantitative metrics and methods may be applied to estimate the effect of harmonization:
(1)First, different statistical criteria may be used to estimate the following endpoints:
(a)Correlation analysis for the gene expression profiles before and after harmonization [29,30,31,33,34];(b)Comparison of between- and within-class distances before and after harmonization [29];
(2)Alternatively, one may classify the samples according to gene expression data after normalization, involving various machine learning (ML) methods:
(a)Logistic regression [92], used in [30];(b)SVM [93], used in [29,31];(c)Nearest shrunken centroids Prediction Analysis for Microarrays (PAM) [94], used in [29].


As a typical material for such normalization quality benchmarks, in many studies, the investigators used standardized reference samples, whose gene expression was interrogated with different equipment using different experimental protocols. Probably, the most important series of such cross-comparisons was performed within the Microarray Quality Control (MAQC) [17,18,19] and Sequencing Quality Control (SEQC) [27] projects mentioned above in this article. 

The MAQC and SEQC projects were focused on profiling the specific model human mRNA sample types. One was the commercial Stratagene universal human reference RNA mixture for all but brain tissues; another one was the commercial Ambion human brain reference mRNA, and the two remaining types were the mixtures of the Stratagene/Ambion samples in the ratios of 3:1 and 1:3, respectively. 

The quality assessment is based on the expectation that a perfect harmonization must support the similarity of gene expression profiles according to the biological nature of the sample rather than depending on the equipment and reagents used to interrogate gene expression. Thus, early approaches used visual inspection of the principal component analysis (PCA) plots and/or cluster dendrograms to assess the cross-platform harmonization benchmarks [30,31,32,33,34]. However, this could only support a manual qualitative assessment without precise quantitative interrogation of the complex class distribution profiles. 

We recently proposed a new metric for the algorithmic cluster analysis of dendrograms [33,95] called Watermelon Multisection (WM). WM measures the strength of data matching with the trait of interest. When moving from the root of the dendrogram to its distal branches, one can calculate general decrease of entropy and, therefore, information gain (IG) at each node of the dendrogram, i.e., its split into two shoulders [95]. This accumulated and normalized IG constitutes the WM metric for a given dendrogram, and a given set of classes under analysis. Consequently, the ratio $=\frac{WM_{S}}{WM_{P}}$, where *WM_S_* is WM metric for clustering according to classes corresponding to biological nature and *WM_P_*, according to the experimental platform used, may be used as a facile yet robust estimate of the harmonization quality. A higher *R* corresponds to a better quality, and vice versa [33].
biomedicines-10-02318-t001_Table 1Table 1Selected benchmarks of harmonization methods.Reference for ComparisonMethodsMaterialsExperimental PlatformQualitative CriteriaQuantitativeCriteriaBest Methods[29]Cross-Platform Normalization, XPN [29]; Column Sample (CS);Median Center (MC); Empirical Bayes (EB) [39];Distance-Weighted Discrimination (DWD) [41,42]Three breast cancer datasets [96,97,98]Affymetrix GeneChip U95Av2 arrays [96];25K Agilent oligonucleotide arrays [97,98]⸺Average distance to nearest sample in anotherplatform; correlation with column standardization data; global integrative correlation; preservation of significantly differential genesXPN[31]XPN; DWD; EB (ComBat) [39]; Median Rank Scores (MRS) [36]; Quantile Discretization (QD) [36]; Normalized Discretization (NorDi) [37]; Distribution Transformation (DisTran) [36,38]; Gene Quantiles(GQ) [43]; Quantile Normalization (QN) [28]MAQC dataset [17,18,19]Human GenomeSurvey Microarrayv2.0;Agilent-012391 Whole Human Genome Oligo Microarray G4112A;Affymetrix Human Genome U133 Plus 2.0 Array;Illumina Sentrix Human-6 Expression Beadchip⸺Mean-mean regression; cross-dataset data transfer for linear SVM [94] and nearest shrunken centroids [95] classificationXPN (for datasets of comparable size);DWD(for datasets of non-comparable size)[30]XPN; DWD; platform-independent latent Dirichlet allocation (PLIDA) [30]Prostate cancer datasets [99,100];Breast cancer datasets [97,101]; MAQCAffymetrix Human Genome U133 Array;Agilent Human 1A (V2);Human GenomeSurvey Microarrayv2.0;Agilent-012391 Whole Human Genome Oligo Microarray G4112A;Affymetrix Human Genome U133 Plus 2.0 Array;Illumina Sentrix Human-6 Expression BeadchipVisual inspection of PCA plots.Correlation analysis between the profiles before and after normalization;cross-dataset data transfer for logistic regression classification [92]PLIDA[67]MatchMixeR (MM) [66]; DWD; XPM;ComBatNCI60 cell lines (dataset 1:58 lines; dataset 2:59 lines)Affymetrix Human Genome U133A array; Human GenomeU133 Plus 2.0 Array;Agilent Human Genome Whole Microarray;Illumina HiSeq 2000⸺R^2^ score (R^2^ is the proportion of the variation in the dependent variable that is predictable from the independent variable [102] analysis;F1 score (F1 score is the harmonic mean of precision and recall [103,104]) analysisMM[32]Shambhala-1; QN; Differential Gene Expression in Sequencing 2 (DESeq2) [59,60,61]MAQC; SEQC datasets [27]Agilent-012391 Whole Human Genome Oligo Microarray G4112A;Affymetrix Human Genome U133 Plus 2.0 Array;Illumina Sentrix Human-6 Expression Beadchip;Illumina HiSeq 2000;Illumina HumanHT-12 V4.0 expression beadchip;Affymetrix Human Gene 2.0 ST Array;Affymetrix GeneChip^®^ PrimeView™ Human Gene Expression ArrayVisual inspection of cluster dendrograms⸺Shambhala-1 (linear Shambhala)[34]CuBlock [34];ComBat [39] YuGene [71]; DBNorm [105]; Shambhala-1 [32]; Universal exPression Code (UPC) [63]MAQCAgilent-012391 Whole Human Genome Oligo Microarray G4112A;Affymetrix Human Genome U133 Plus 2.0 Array;Illumina Sentrix Human-6 Expression BeadchipVisual inspection of cluster dendrograms and PCA plotsCross-dataset data transfer for support vector machine (SVM) classification [93]CuBlock[33]Shambhala-2; Shambhala-1; QN; DESeq2; CuBlock; robust QN (QNR) [91]; Training Distribution Machine (TDM) [62]; UPCGTEx [11], The Cancer Genome Atlas (TCGA) [10]; Oncobox Atlas of Normal Tissue Expression (ANTE) [13]; MAQC; SEQCIllumina HiSeq 2000;Illumina HiSeq 3000;Agilent-012391 Whole Human Genome Oligo Microarray G4112A;Affymetrix Human Genome U133 Plus 2.0 Array;Illumina Sentrix Human-6 Expression Beadchip;Illumina HumanHT-12 V4.0 expression beadchip;Affymetrix Human Gene 2.0 ST Array;Affymetrix GeneChip^®^ PrimeView™ Human Gene Expression ArrayVisual inspection of PCA plotsWatermelon Multisection metric for quantitative assessment of clustering on dendrograms [95]Shambhala-2 (cubic Shambhala)


## 4. Application Notes

During the last two decades, quantile normalization (QN) [28] and differential gene expression in sequencing 2, DESeq2 [59,60,61], have become methods of choice for intra-platform normalization of the MH and RNAseq gene expression data, respectively. 

However, for the cross-platform harmonization, dozens of methods were developed for both MH and NGS types of gene expression data, but none of them was so far recognized as the gold standard. In fact, many, if not most, aspects of intra- and cross-platform normalization, such as incomparability of profiles obtained using different platforms, numerous methods for cross-platform normalization, and performance benchmarks for them, were studied in the first decade of the XXI century, in the co-called MH era. The advent of NGS, however, did not make this problem unimportant, at least because there is still a problem regarding how to harmonize old MH and new NGS data. 

Most of cross-platform normalization methods return the output data in the flexible format, which requires recalculation of all previously processed profiles when adding new data to the analysis. This may dramatically increase calculation time and costs which can grow exponentially with the increase of the sample size. Furthermore, some methods which show the best performance in cross-platform normalization tests [31], such as XPN [29], have serious limitations. For instance, XPN allows normalization of only two datasets at once, with no subsequent application to other datasets [29]. In addition, an unbalanced size of groups of samples under harmonization may create obstacle to the analysis of the whole groups. The latter may force researchers to arbitrarily decrease samplings and not to include all available data into the analysis. 

Thus, the need for the predefined, uniformly-shaped output for data harmonization was formulated about a decade ago [62,63,79,80,81,82]. Recently, we proposed a concept of uniformly shaped cross-platform harmonization of gene expression profiles [32,33]. The key feature of such harmonization is that each profile is converted into the shape of the reference definitive dataset independently from other profiles under harmonization. In such a way, the unlimited number of datasets of any size each can be harmonized. Furthermore, adding new data to the analysis does not require recalculation for the previously harmonized profiles, which spares time and reduces costs. 

With such a concept in mind, we obtained the best results with the cubic data transformation algorithm adopted from the CuBlock method [34] and built Shambhala-2 package [33]. Shambhala-2 showed a strong capacity to restore the correct order of clusters on dendrograms, when the samples were grouped according to their biological nature, not the technical platform used to profile gene expression. This was effective for both MH and RNAseq types of data, including mixed MH-RNAseq datasets under harmonization. We hope, therefore, that this generation or next generations of Shambhala harmonizer will find their niche in the analysis of big transcriptomic data in the future. 

## 5. Conclusions

We summarized our experience of using various harmonization methods for gene expression profiles in Table 2. 

For the intra-platform harmonization of the MH data, QN [28] may seem the method of choice; however, the “robust QN” (QNR) [91] showed generally worse performance than the ordinary QN [33]. In turn, for the intra-platform harmonization of the NGS data, DESeq2 method could be recommended [59,60,61]. 

In case of cross-platform harmonization of two datasets with a comparable number of gene expression profiles, the best choice could be the XPN method [29,31]. When the data are MH profiles and there are more than two datasets under analysis, the method CuBlock [34] is preferred.

Finally, in the case of merging the MH and NGS expression datasets, or when merging of data from various platforms is needed, and the uniformly shaped (suitable for further intercomparisons) output format is required, then Shambhala-1 [32] or Shahmhala-2 [33] technique can be the best option. 

To our knowledge, Shambhala methods were the first gene expression harmonizers with a uniformly shaped output, which were applied to merge the RNAseq and MH profiles [32,33]. Thus, these methods may become useful for the broad spectrum of applications. 

However, one should keep in mind that Shambhala-1/2 approaches are algorithmically complex and, therefore, computational resource-demanding. Thus, parallel execution of the program code may be advantageous [33,89].
biomedicines-10-02318-t002_Table 2Table 2Recommendations for the use of selected intra- and cross-platform harmonization methods.ReferenceMethod Mathematical Principle Algorithmic Complexity AdvantagesShortcomings[28]Quantile normalization (QN) Ranking the expression levels of different genes within each profile and setting the expression level of each gene to the mean value (over all profiles) for the respective rank Relatively simpleGold standard method for intra-platform normalization of the MH dataAvoiding being used for cross-platform harmonization of the MH data;requiring recalculation of all gene expression-based values after addition of new samples [59,60,61]Differential Gene Expression in Sequencing 2 (DESeq2)Transform based on the negative binomial distribution Moderately complexGold standard for intra-platform normalization of RNAseq dataRequiring recalculation of all gene expression-based values after addition of new samples[29]Cross-Platform Normalization (XPN) Piecewise linear iterative transformRelatively complexThe method of choice for harmonization of two datasets of comparable sizeAllowing normalization of more than two datasets; not recommending subsequent application to other datasets; requiring recalculation of all gene expression-based values after addition of new samples[34]CuBlockPiecewise cubic iterative transformRelatively complexThe method of choice for cross-platform normalization of more than two MH datasetsRequiring recalculation of all gene expression-based values after addition of new samples[32]Shambhala-1 (linear Shambhala)Uniformly shaped harmonization based on the XPN method.ComplexWorking for harmonization of unlimited number of datasets of any size, for both MH and RNAseq data or their combinations; not requiring recalculation of gene expression-based values after addition of new samplesResource-demanding[33]Shambhala-2 (cubic Shambhala)Uniformly shaped harmonization based on the CuBlock method.ComplexWorking for harmonization of the unlimited number of datasets of any size, for both MH and RNAseq data or their combinations; not requiring recalculation of gene expression-based values after addition of new samples Resource-demanding


## Data Availability

Not applicable.

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
