# Peer review of "Transcriptomic Harmonization as the Way for Suppressing Cross-Platform Bias and Batch Effect"

_biomedicines, 2022, doi:10.3390/biomedicines10092318_

Round 1

Reviewer 1 Report

Comments on the manuscript:

“Transcriptomic harmonization as the way for suppressing cross-platform bias and batch effect”

The emergence of high-throughput methods for detecting gene expression is where quantitative transcriptomics originated. However, this requires the comparison of expression profiles obtained with different equipment and protocols. Many variables act on gene expression signals, which translates into a large or even plethoric number of specific characteristics of the transcriptomic profiles deposited in databases. Their use requires comparisons, which sometimes makes their use tricky. The study presented here concerns an overview as broad as possible of the transcriptomic harmonization approaches and methods currently used and their application.

This article reviews powerful high-throughput methods for detecting gene expression. It seems to me to be useful at a time when these increasingly used methods are numerous and give variable results from one method to another and which therefore need to be compared, which is also the purpose of this work. This article could be published after however some improvements of the manuscript. The text seems to me globally well documented and correctly written. The use of tables is useful and avoids overloading an already large text.

I will make only a few minor remarks.

A short introduction specifying the purpose of the study would seem useful to me. Not being a specialist in these methods, it would seem useful to me to indicate how they were chosen to be the subject of this work.

Page 1, line 39: “GEO” is not in the list of acronyms

Page 1; line 43: write: “Atlas of Normal Tissue Expression (ANTE)”.

Page 2, lines 89-93: write “Quantile Discretization (QD)”, “Normalized Discretization (NorDi)”, “Distribution Transformation (DisTran)”, Empirical Bayes (EB)”, “Gene Quantiles GQ”.

Page 2, line 91: add “ComBat” in the list of acronyms.

Page 2, line 95: next-generation sequencing (NGS) is not in the list of acronym.

Page 3, line 147: “MBEI” is not in the list of acronyms.

Page 3, line 148: “PLIER” is not in the list of acronyms.

Page 7, table 1: “CuBlock” and “QNR” are not in the list of acronyms.

Check the bibliographic references

Page 10, line 355, reference 10. “Czerwi?ska, P.”: check the name.

Author Response

Reviewer 1:

-The emergence of high-throughput methods for detecting gene expression is where quantitative transcriptomics originated. However, this requires the comparison of expression profiles obtained with different equipment and protocols. Many variables act on gene expression signals, which translates into a large or even plethoric number of specific characteristics of the transcriptomic profiles deposited in databases. Their use requires comparisons, which sometimes makes their use tricky. The study presented here concerns an overview as broad as possible of the transcriptomic harmonization approaches and methods currently used and their application.

This article reviews powerful high-throughput methods for detecting gene expression. It seems to me to be useful at a time when these increasingly used methods are numerous and give variable results from one method to another and which therefore need to be compared, which is also the purpose of this work. This article could be published after however some improvements of the manuscript. The text seems to me globally well documented and correctly written. The use of tables is useful and avoids overloading an already large text.

I will make only a few minor remarks.

A short introduction specifying the purpose of the study would seem useful to me. Not being a specialist in these methods, it would seem useful to me to indicate how they were chosen to be the subject of this work.

>Response: we added an ending paragraph to the introductory section:

“In this review we classify available intra- and cross-platform harmonization methods of transcriptomic profiles and compare their performance characteristics. Finally, we also included practical recommendations that may guide the reader to select optimal method depending on a specific task.”    

-Page 1, line 39: “GEO” is not in the list of acronyms

>Response: term added to the abbreviations list.

-Page 1; line 43: write: “Atlas of Normal Tissue Expression (ANTE)”.

>Response: done.

-Page 2, lines 89-93: write “Quantile Discretization (QD)”, “Normalized Discretization (NorDi)”, “Distribution Transformation (DisTran)”, Empirical Bayes (EB)”, “Gene Quantiles GQ”.

>Response: done.

-Page 2, line 91: add “ComBat” in the list of acronyms.

>Response: done.

-Page 2, line 95: next-generation sequencing (NGS) is not in the list of acronyms.

>Response: term added to the list in the revised manuscript.

-Page 3, line 147: “MBEI” is not in the list of acronyms.

>Response: term added to the list in the revised manuscript.

-Page 3, line 148: “PLIER” is not in the list of acronyms.

>Response: term added to the list in the revised manuscript.

-Page 7, table 1: “CuBlock” and “QNR” are not in the list of acronyms.

>Response: term added to the list in the revised manuscript.

-Check the bibliographic references

Page 10, line 355, reference 10. “Czerwi?ska, P.”: check the name

>Response: fixed in the revised version

Reviewer 2 Report

The description in Table 1 seems biased in terms of materials employed for data sets.

I have following comments.

Conclusions style derails from the standard. Summarized comments should be provided in addition to the table.

More than 2/3 of references are published before 2015. The authors had better update majority of them.   

Author Response

Reviewer 2:

-The description in Table 1 seems biased in terms of materials employed for data sets.

>Response:

Most of the performance indicators for harmonization techniques were interrogated with the Stratagene and Ambion reference expression profiles used for the MAQC and SEQC projects, but we now added another recent control sample type based on the NCI60 cell line RNA (reference [67]).

-I have following comments.

Conclusions style derails from the standard. Summarized comments should be provided in addition to the table.

>Response: we added the following paragraphs to Conclusions:

“For the intra-platform harmonization of the MH data, QN [28] may seem the method of choice; however, the “robust QN” (QNR) [92] showed generally worse performance than the ordinary QN [33]. In turn, for the intra-platform harmonization of the NGS data, DESeq2 method could be recommended [60–62].  

In case of cross-platform harmonization of two datasets with comparable number of gene expression profiles, the best choice could be the XPN method [29,31]. When the data are MH profiles, and there are more than two datasets under analysis, the method CuBlock [34] is preferred.

Finally, in the case of merging the MH and NGS expression datasets, or when merging of data from various platforms is needed, and the uniformly shaped (suitable for further intercomparisons) output format is required, then Shambhala-1 [32] or Shahmhala-2 [33] technique can be the best option.

To our knowledge, Shambhala methods were the first gene expression harmonizers with the uniformly shaped output, which were applied to merge the RNAseq and MH profiles [32,33]. Thus, these methods may become useful for the broad spectrum of applications.

However, one should keep in mind that Shambhala-1/2 approaches are algorithmically complex and, therefore, computational resource-demanding. Thus, parallel execution of the program code may be advantageous [33,90].”

-More than 2/3 of references are published before 2015. The authors had better update majority of them. 

>Response: indeed, many aspects of intra- and cross-platform harmonization were studied in the first decade of the XXI century, in the “microarray era”. Emergence of NGS techniques, however, did not eliminate this problem, at least because there is still exist a problem how to harmonize the previous MH with the new NGS data. To provide a more recent-oriented scope, we added fifteen new references, thus increasing citation of papers published after 2015 to 43%.

Reviewer 3 Report

Dear Authors,

The review article speaks about the emerging transcriptomic harmonization or methods used for gene expression. This is updated and collected information about gene sequencing and related diseases and disorders. I hope this information is beneficial to the research in gene expression studies. These review reports are collective information from high-throughput techniques. Numerous factors influence gene expression signals, resulting in a vast or even plethoric number of distinct features of the transcriptome profiles stored in databases. This paper provides a comprehensive review of the existing methodologies and methods for transcriptome harmonization and their applications. This article discusses efficient and current high-throughput techniques for detecting gene expression. It appears beneficial at a time when these increasingly used procedures are many and provide outcomes that vary from method to method, necessitating comparison, which is also the objective of this study.

Author Response

>We are very thankful to the Reviewer 3 for his\her positive feedback.

Round 2

Reviewer 2 Report

The manuscript has been revised accordingly.